*Research Directions:*
*Depression*

## Results

depression; prediction; fMRI; prognosis; difficult-to-treat

**Corresponding author:**
Roland Zahn; Email: roland.zahn@kcl.ac.uk

# Neural signatures of emotional biases predict clinical outcomes in difficult-to-treat depression

Diede Fennema[1] ⓘ, Gareth J. Barker[2] ⓘ, Owen O'Daly[2] ⓘ, Beata R. Godlewska[3,4] ⓘ, Ewan Carr[5] ⓘ, Kimberley Goldsmith[5] ⓘ, Allan H. Young[1,6] ⓘ, Jorge Moll[7] ⓘ and Roland Zahn[1,6,7] ⓘ

[1]Centre of Affective Disorders, Institute of Psychiatry, Psychology & Neuroscience, King's College London, London, UK; [2]Department of Neuroimaging, Institute of Psychiatry, Psychology & Neuroscience, King's College London, London, UK; [3]Psychopharmacology Research Unit, University Department of Psychiatry, University of Oxford, Oxford, UK; [4]Oxford Health NHS Foundation Trust, Warneford Hospital, Oxford, UK; [5]Department of Biostatistics and Health Informatics, Institute of Psychiatry, Psychology & Neuroscience, King's College London, London, UK; [6]National Service for Affective Disorders, South London and Maudsley NHS Foundation Trust, London, UK and [7]Cognitive and Behavioral Neuroscience Unit, D'Or Institute for Research and Education (IDOR), Pioneer Science Program, Rio de Janeiro, Brazil

## Abstract

**Background:** Neural predictors underlying variability in depression outcomes are poorly understood. Functional MRI measures of subgenual cortex connectivity, self-blaming and negative perceptual biases have shown prognostic potential in treatment-naïve, medication-free and fully remitting forms of major depressive disorder (MDD). However, their role in more chronic, difficult-to-treat forms of MDD is unknown.

**Methods:** Forty-five participants (n = 38 meeting minimum data quality thresholds) fulfilled criteria for difficult-to-treat MDD. Clinical outcome was determined by computing percentage change at follow-up from baseline (four months) on the self-reported Quick Inventory of Depressive Symptomatology (16-item). Baseline measures included self-blame-selective connectivity of the right superior anterior temporal lobe with an *a priori* Brodmann Area 25 region-of-interest, blood-oxygen-level-dependent *a priori* bilateral amygdala activation for subliminal sad vs happy faces, and resting-state connectivity of the subgenual cortex with an *a priori* defined ventrolateral prefrontal cortex/insula region-of-interest.

**Findings:** A linear regression model showed that baseline severity of depressive symptoms explained 3% of the variance in outcomes at follow-up ($F[3,34] = .33$, $p = .81$). In contrast, our three pre-registered neural measures combined, explained 32% of the variance in clinical outcomes ($F[4,33] = 3.86$, $p = .01$).

**Conclusion:** These findings corroborate the pathophysiological relevance of neural signatures of emotional biases and their potential as predictors of outcomes in difficult-to-treat depression.

## Introduction

Currently, treatment of major depressive disorder (MDD) is based on a trial-and-error approach, with only half of patients responding to their initial treatment (Rush et al. 2006). There is a clear need for improving treatment in patients with depression, which could be informed by standard clinical variables and biomarkers (Dunlop and Mayberg 2014; Fonseka et al. 2018; Perlman et al. 2019). The field has started to identify various biomarkers showing promise, such as genetic markers (Breitenstein et al. 2014; Laje et al. 2009), behavioural and cognitive markers (Groves et al. 2018; Park et al. 2018; Perna et al. 2020), metabolic and inflammatory markers (Lopresti et al. 2014; Schmidt et al. 2011) and neuroimaging markers (Breitenstein et al. 2014; Dichter et al. 2015; Dunlop and Mayberg 2014; Fonseka et al. 2018; Fu et al. 2013).

Such biomarkers are thought to represent underlying biological substrates of depression, which can be used to predict general prognosis regardless of treatment, better outcome with any treatment, or differential treatment response (Simon and Perlis 2010). For example, baseline metabolic profile was found to differentiate between responders and non-responders to sertraline or placebo (Kaddurah-Daouk et al. 2011), baseline C-reactive protein differentially predicted response to escitalopram or nortriptyline (Uher et al. 2014), and baseline resting-state functional connectivity with the subgenual cortex differentially predicted response to antidepressant treatment or cognitive behavioural therapy (Dunlop et al. 2017).

It is important to note, however, that MDD is a multifaceted disorder associated with a wide range of cognitive, behavioural, emotional and physiological symptoms (Disner et al. 2011). As such, it is unlikely that a single clinical or biological marker can predict treatment outcome

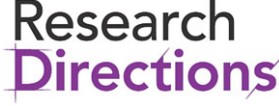

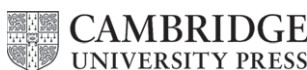

(Patel et al. 2016; Phillips et al. 2015). In fact, Lee et al. (2018) showed that models informed by multiple data types, such as a composite of clinical features, neuroimaging, or genetic measures, were more accurate at predicting outcome than less complex models. Nonetheless, current clinical practice is mostly based on questionnaire- and interview-based assessments, which represent a wealth of clinical data which can be used to predict treatment outcome (Rost et al. 2022).

In recent years, machine-learning methods have been increasingly employed to examine which clinical variables are most predictive of response or remission, allowing identification of patterns of information at an individual patient level (Chekroud et al. 2021; Jankowsky et al. 2024). Various studies have consistently identified baseline symptom severity, number of depressive episodes and co-morbid anxiety disorders as predictors of treatment outcome (Balestri et al. 2016; Bartova et al. 2019; Chekroud et al. 2016; Iniesta et al. 2016; Kautzky et al. 2018; Perlis 2013). However, standard clinical variables alone capture a limited amount of variance in clinical outcome, with estimates in the region of 5–10% (Iniesta et al. 2016; Perlis 2013), and they tend to perform worse than neuroimaging measures (Dunlop 2015; Jollans and Whelan 2016; Lee et al. 2018; Poirot et al. 2024; Schmaal et al. 2015).

Neuroimaging measures may be of particular interest, as dysfunctional neural processes are core to the development and maintenance of depressive symptoms (Godlewska 2020). They capture emotional biases associated with depression, such as the tendency to focus more on negative facial expressions than positive ones (Bourke et al. 2010; Krause et al. 2021), proneness to experience excessive self-blaming emotions, such as overgeneralised guilt and disgust/contempt towards oneself (Duan et al. 2021; Duan et al. 2023; Green et al. 2013; Weiner 1985; Zahn et al. 2015), as well as rumination, i.e., a tendency to engage in recursive, automatic thoughts often linked to self-critical thinking (Berman et al. 2014; Hamilton et al. 2015; Nolen-Hoeksema et al. 2008).

Leading neuroanatomical models of MDD propose that impaired function within prefrontal-limbic neural circuits, particularly the subgenual cingulate cortex and amygdala, explains disruptions of emotional processing and regulation associated with depression (Price and Drevets 2010; Ressler and Mayberg 2007). Neuroimaging biomarkers capturing the aforementioned – often implicit – emotional biases associated with depression have shown promise in predicting prognosis in MDD at an individual level, notably amygdala activation in response to emotional faces (Williams et al. 2015) and resting-state posterior subgenual cortex connectivity (Dunlop et al. 2017) in current MDD, and self-blame-selective anterior temporal-subgenual connectivity in remitted MDD (Lawrence et al. 2022). Despite these promising findings, studies tend to focus on treatment-naïve and treatment-free samples of MDD, and it is unclear whether these neural signatures generalise to pragmatic samples of patients encountered in clinical settings. Moreover, it is important to establish whether imaging measures provide added value in predicting clinical outcomes compared to standard baseline clinical variables.

Here, we probed the potential of these neural signatures of emotional biases in predicting clinical outcomes in a pragmatic sample of difficult-to-treat MDD after four months of primary care. These pre-registered (NCT04342299) neural signatures were selected based on their potential to predict response to treatment at an individual level and cover complementary neurocognitive aspects of MDD, i.e., self-blaming biases, negative perceptual biases and dysfunction of task-independent subgenual networks.

## Methods

The functional MRI (fMRI) dataset reported here was collected as part of an observational sub-study within a feasibility trial, the Antidepressant Advisor Study (NCT03628027) (Harrison et al. 2020; Harrison et al. 2022). We have published tasked-based functional imaging (Fennema et al. 2023; Fennema, Barker, O'Daly, Duan, Godlewska, et al. 2024) and resting-state fMRI results (Fennema, Barker, O'Daly, Duan, Carr, et al. 2024) from the same cohort previously, but here, we report on the prediction model for the first time.

### Participants

Forty-five participants fulfilled criteria for current MDD according to the Diagnostic and Statistical Manual of Mental Health Disorders, Fifth Edition (DSM-5) (First et al. 2015) and had not responded to at least two serotonergic antidepressants. Participants were encouraged to book an appointment with their general practitioner (GP) to review their medication and followed up after receiving four months of standard care. For more information about inclusion/exclusion criteria, recruitment and assessment, please see Supplementary Methods.

Prior to their medication review, participants attended an fMRI session, consisting of three paradigms: the moral sentiment task (assessing self-blame-related biases), the subliminal faces task (assessing bias in emotional processing), and a resting-state fMRI scan. As part of the moral sentiment task, participants viewed self- and other-blaming emotion-evoking statements. Participants were shown written statements describing actions counter to socio-moral values described by social concepts (e.g., impatient, dishonest) in which the agent was either the participant (self-agency) or their best friend (other-agency) (Fennema et al. 2023). As part of the subliminal faces task, participants were presented with a series of faces. The faces were shown in pairs, briefly displaying a "target" face (expressing sad, happy or neutral emotion) followed by another "mask" face of neutral expression (Fennema, Barker, O'Daly, Duan, Godlewska, et al., 2024). As part of the resting-state fMRI scan, participants were instructed to keep their eyes open and let their mind wander while focusing on a cross (Fennema, Barker, O'Daly, Duan, Carr, et al., 2024). For more details on the fMRI paradigms, please see Supplementary Materials.

### Main outcome

As stated in our pre-registered protocol (NCT04342299), we used a continuous measure of clinical outcome rather than categorising participants into responders and non-responders using the standard definition of a 50% reduction (Nierenberg and DeCecco 2001) in self-reported Quick Inventory of Depressive Symptomatology (16-item; QIDS-SR16) (Rush et al. 2003) scores, due to an unbalanced split between the resulting groups (responders n = 8; non-responders n = 30). The outcome was defined as the percentage change at follow-up from baseline on our pre-registered primary outcome measure, QIDS-SR16, where negative scores corresponded to a reduction in depressive symptoms.

### fMRI measures

Statistical Parametric Mapping 12 was used for blood-oxygen-level-dependent (BOLD) effect analysis and psychophysiological interaction analysis, while Data Processing Assistant for Resting-

State fMRI was used for resting-state analysis (please see Supplementary Methods for more details). Regression coefficient averages (moral sentiment task and subliminal faces task) and cluster mean z-score (resting-state scan) over our pre-registered regions-of-interest (ROIs) were extracted for individual participants using the MarsBaR toolbox (Rorden and Brett 2000), i.e., self-blame-selective connectivity between the right superior anterior temporal lobe (RSATL) and posterior subgenual cortex (Brodmann Area [BA] 25), bilateral amygdala BOLD activation for subliminal sad vs happy faces, and resting-state functional connectivity between the bilateral posterior subgenual cortex and left ventrolateral prefrontal cortex (BA47; ventrolateral prefrontal cortex [VLPFC])/insula. For more details, please see Supplementary Materials.

## Statistical analysis

Multiple linear regression was used to assess potential predictors of QIDS-SR16 percentage change, as well as an exploratory logistic regression to determine likelihood of response vs. non-response. The aim of the study was to estimate the effect size of using our pre-registered imaging measures as predictors of clinical outcomes, rather than tease out the importance of each predictor given the limitations of our sample size. As such, we ran our main "fMRI" multivariable model which assessed the contribution of our three pre-registered fMRI measures as outlined above, with baseline Maudsley Modified Patient Health Questionnaire, 9 items (MM-PHQ-9; measure of severity of depressive symptoms) (Harrison et al. 2021) as a covariate.

In addition, we ran a supplementary "clinical" multivariable model to compare the contribution of standard clinical measures (baseline MM-PHQ-9, baseline Generalised Anxiety Disorder, 7-items (GAD-7; measure of severity of anxiety symptoms) (Spitzer et al. 2006), and Maudsley Staging Method total score (proxy of treatment-resistance based on duration, severity and treatment failures) (Fekadu et al. 2018); please see Supplementary Methods for more details on the clinical measures). Another supplementary "high-quality fMRI" multivariable model assessed the impact of suboptimal fMRI quality, i.e., signal drop-out and/or more motion, on the predictive value of the fMRI measures, including only participants with high-quality fMRI data for all three scans (n=30). Other supplementary models considered the individual contribution of the pre-registered fMRI measures (please see Supplementary Methods and Results).

Please note that our pre-registered imaging measures also included additional regions of interest: functional resting-state subgenual cortex connectivity with the left ventromedial prefrontal cortex (BA10) and with the dorsal midbrain (Fennema, Barker, O'Daly, Duan, Carr, et al. 2024), as well as pregenual anterior cingulate cortex BOLD activation for subliminal sad vs happy faces (Fennema, Barker, O'Daly, Duan, Godlewska, et al. 2024). However, as our sample size only allowed us to model a limited number of variables without risk of overfitting, for our primary prediction model, we solely included variables showing univariate prediction effects in our previous analyses (Fennema, Barker, O'Daly, Duan, Carr, et al. 2024; Fennema, Barker, O'Daly, Duan, Godlewska, et al., 2024). For more details on the exploratory "pre-registration" model, please see Supplementary Methods.

All variables were Fisher $Z$-transformed to derive beta coefficients and corresponding standard error. Correlation analysis (Spearman's rho) was used to investigate the association between the pre-registered neural signatures. To test whether there is any link between treatment change and symptom change, a one-way analysis of variance was conducted (please see Supplementary Methods for a description of treatment change). All tests were carried out using IBM SPSS Statistics 27, using a significance threshold of $p = .05$, two-tailed.

## Results

### Subgroup characteristics

Table 1 presents participant characteristics at baseline, split by responders and non-responders. Of 45 included participants, 38 had usable fMRI data (31 [82%] female, mean [SD] age = 41.8 [14.8] years). Most participants fulfilled the DSM-5 anxious distress specifier criteria (82%) and met criteria for a life-time axis I co-morbidity (87%). Average baseline depression severity was severe according to MM-PHQ-9 (mean [SD] = 18.7 [4.7]) and QIDS-SR16 (mean [SD] = 17.3 [3.5]), and 82% of the participants were taking a selective serotonin-reuptake inhibitor. There were no significant differences between responders and non-responders at baseline in terms of demographic and clinical characteristics ($t < 1.31$ and $p > .20$), except for current major depressive episode duration (responders mean [SD] = 6.3 [5.3]; non-responders mean [SD] = 32.8 [50.2]; $t[31.2] = -2.85$, $p = .01$).

As part of the study, participants were encouraged to book an appointment with their GP to review their antidepressant medication. Even though UK care guidelines would recommend changing antidepressant medications in non-responders, unexpectedly, more than half (55%) did not change their medication and some even stopped their medication (16%; Supplementary Table 1). Despite little change in treatment, on average, participants showed a significant reduction in depressive symptoms from baseline to follow-up in QIDS-SR16 scores (mean [95% CI] = −4.1 [−5.8, −2.4]). This was also the case for other self- and observer-rated scores (Supplementary Table 2).

There was a mean percentage change [SD] of −23.1 [30.0] in QIDS-SR16: those with a relevant change showed the most improvement in QIDS-SR16 (mean percentage change [SD] = −43.8 [20.3]), followed by participants with a minimal change (mean percentage change [SD] = −32.1 [32.4]) and participants with no change (mean percentage change [SD] = −17.6 [29.6]). However, there was no significant difference between the groups ($F[2,37] = 1.78$, $p = .18$).

### Prediction models

The "fMRI" model using the pre-registered fMRI measures with baseline MM-PHQ-9 as a covariate explained 32% of the variance of QIDS-SR16 percentage change ($F[4,33] = 3.86$, $p = .01$, $R^2 = .32$, $R^2_{adjusted} = .24$; Table 2). When including all previously pre-registered regions, the overall prediction effect for the "pre-registration" model was comparable ($R^2 = 33\%$, please see Supplementary Results). When limiting to "high-quality fMRI," the model explained 43% of the variance of QIDS-SR16 percentage change ($F[4,25] = 4.67$, $p = .01$, $R^2 = .43$, $R^2_{adjusted} = .34$; Supplementary Table 3). In contrast, the "clinical" model using standard clinical measures at baseline, i.e. MM-PHQ-9, GAD-7 and Maudsley Staging Method, explained only 3% of the variance of QIDS-SR16 percentage change ($F[3,34] = .33$, $p = .81$, $R^2 = .03$, $R^2_{adjusted} = -.06$; Table 2).

Bilateral amygdala BOLD activation positively contributed to the variance in QIDS-SR16 percentage change (partial $\beta = 11.11$, $t[33] = 2.21$), while partial effects of resting-state functional

**Table 1.** Baseline demographic and clinical characteristics by responders and non-responders (n = 38)

| Characteristic | Responders (n = 8) | Non-responders (n = 30) |
| --- | --- | --- |
| | n (%) or mean ± SD; range | |
| **Age, in years** | 42.9 ± 16.1; 19-66 | 41.6 ± 14.8; 20-62 |
| **Sex** | | |
| Female | 7 (88) | 24 (80) |
| Male | 1 (13) | 5 (17) |
| Other | 0 (0) | 1 (3) |
| **Ethnicity[a]** | | |
| Asian | 1 (13) | 24 (80) |
| Black | 0 (0) | 2 (7) |
| White | 5 (63) | 3 (10) |
| Other ethnicity | 1 (13) | 1 (3) |
| **Years of education, in years** | 17.4 ± 3.3; 12-22 | 16.9 ± 3.6; 10-24 |
| **Depression severity** | | |
| Current MDE duration, in months | 6.3 ± 5.3; 1-15 | 32.8 ± 50.2; 1-176 |
| Number of MDEs | 7.3 ± 5.6; 3-20 | 6.2 ± 4.8; 1-20 |
| MM-PHQ-9 total score | 20.0 ± 3.8; 13-25 | 18.4 ± 4.9; 8-27 |
| QIDS-SR16 total score | 17.9 ± 3.9; 11-22 | 17.2 ± 3.5; 10-23 |
| MADRS total score | 29.5 ± 5.0; 23-38 | 32.0 ± 4.9; 22-42 |
| SOFAS total score | 55.9 ± 3.5; 52-61 | 53.1 ± 5.7; 33-61 |
| **Maudsley Staging Method** | | |
| Mild | 3 (38) | 12 (40) |
| Moderate | 5 (63) | 18 (60) |
| Severe | 0 (0) | 0 (0) |
| **MDD DSM-5 subtype** | | |
| Anxious distress only | 0 (0) | 7 (23) |
| Melancholic features + anxious distress | 1 (13) | 4 (13) |
| Atypical features only | 0 (0) | 1 (3) |
| Atypical features + anxious distress | 3 (38) | 15 (50) |
| No specific subtype | 4 (50) | 3 (10) |
| **Treatment at baseline** | | |
| SSRI | 6 (75) | 25 (83) |
| SNRI | 1 (13) | 3 (10) |
| Other class | 1 (13) | 2 (7) |
| Non-pharmacological treatment | 4 (50) | 6 (20) |
| **GAD-7 total score** | 10.4 ± 6.6; 1-21 | 11.6 ± 3.6; 5-20 |
| **Life-time axis-I co-morbidity** | | |
| Posttraumatic stress disorder | 2 (25) | 15 (50) |
| Other anxiety disorder | 4 (50) | 12 (40) |
| Obsessive-compulsive disorder | 0 (0) | 3 (10) |
| Eating disorder | 3 (38) | 10 (33) |
| None | 2 (25) | 3 (10) |

Percentages may not add up to 100 due to rounding. MDD = major depressive disorder; DSM-5 = Diagnostic and Statistical Manual for Mental Disorders 5[th] edition; MDE = major depressive episode; SD = standard deviation; MM-PHQ-9 = Maudsley Modified Patient Health Questionnaire, 9 items; QIDS-SR16 = Quick Inventory Depressive Symptomatology, self-rated, 16 items; MADRS = Montgomery-Åsberg Depression Rating Scale; SOFAS = Social and Occupational Functioning Scale; SSRI = selective serotonin reuptake inhibitor; SNRI = selective norepinephrine reuptake inhibitor; GAD-7 = Generalised Anxiety Disorder, 7 items.
[a]Missing data for one participant. Ethnicity categories have been combined: "White" includes White: British, Other and Polish; "Asian" includes Asian or Asian British: Indian, Chinese and Other Asian; "Black" includes Black or Black British: Caribbean.

**Table 2.** Prediction models of clinical outcomes in depression (n = 38)

| | Model parameters | | | | Overall model | |
|---|---|---|---|---|---|---|
| | $\beta$ | SE | t | p | $R^2$ | p |
| **Standard clinical variables model** | | | | | .03 | .81 |
| Baseline MM-PHQ-9 | .15 | 10.86 | .01 | .99 | | |
| Baseline GAD-7 | 3.76 | 8.53 | .44 | .66 | | |
| Maudsley Staging Method | 3.37 | 5.35 | .63 | .53 | | |
| **fMRI measures model** | | | | | .32 | .01* |
| Baseline MM-PHQ-9 | 4.19 | 6.97 | .60 | .55 | | |
| Self-blame-selective RSATL-BA25 connectivity | −7.28 | 4.25 | −1.71 | .10 | | |
| Bilateral amygdala BOLD activation for sad vs happy subliminal faces | 11.11 | 5.04 | 2.21 | .04* | | |
| Resting-state posterior subgenual cortex-VLPFC/insula functional connectivity | −8.15 | 4.18 | −1.95 | .06 | | |

* Significant at $p < .05$ threshold, two-tailed. SE = standard error; MM-PHQ-9 = Maudsley Modified Patient Health Questionnaire, 9 items; GAD-7 = Generalised Anxiety Disorder, 7 items; RSATL = right superior anterior temporal lobe; BA = Brodmann Area; BOLD = blood-oxygen level-dependent; VLPFC = ventrolateral prefrontal cortex.

connectivity between the posterior subgenual cortex and left VLPFC/insula as well as self-blame-selective RSATL-BA25 connectivity contributed negatively (resting-state: partial $\beta = −8.15$, $t[33] = −1.95$; RSATL-BA25: partial $\beta = −7.28$, $t[33] = −1.71$; Figure 1). Please see Supplementary Results and Supplementary Table 3 for exploratory separate prediction models for each fMRI paradigm showing a maximum of 18% variance in clinical outcomes explained, when using the bilateral amygdala BOLD signature.

Notably, there were no bivariate associations between the three pre-registered fMRI measures (self-blame-selective RSATL-BA25 connectivity and bilateral amygdala BOLD activation: $r_s[38] = −.06$, $p = .71$; self-blame-selective RSATL-BA25 connectivity and resting-state functional connectivity between posterior subgenual cortex and left VLPFC/insula: $r_s[38] = .09$, $p = .61$; bilateral amygdala BOLD activation and resting-state functional connectivity between posterior subgenual cortex and left VLPFC/insula: $r_s[38] = −.09$, $p = .61$).

### Exploratory findings responders vs. non-responders

A logistic regression was performed to determine the effects of the pre-registered neural measures and baseline MM-PHQ-9 on the likelihood of response vs. non-response. The logistic regression model was statistically significant, $\chi^2(4) = 11.09$, $p = .03$. The model explained 39% (Nagelkerke $R^2$) of the variance in responders and correctly classified 81.6% of the cases. Increased functional connectivity between the bilateral subgenual cortex and left VLPFC/insula was associated with an increased

likelihood of response. For more details, please see Supplementary Results.

### Discussion

To our knowledge, this is the first study to combine complementary functional imaging measures of affective circuits in MDD and to probe their role in prospectively predicting clinical outcomes in a pragmatic setting. We show that neuroimaging markers hold promise: the model with the three pre-registered fMRI measures explained more variance in clinical outcomes compared with the clinical model, i.e. 32% vs 3%. The model that only included participants with high-quality fMRI measures explained an even larger amount of variance (43%), highlighting the need to adequately account for signal drop-out and/or motion artefacts. However, it is important to acknowledge that no formal statistical tests were undertaken to compare the regression models as the study was not powered for such comparisons, which limits the interpretability of differences between the models.

Interestingly, the effects of the three pre-registered fMRI measures were uncorrelated, showing that these may capture distinct aspects of MDD pathophysiology, i.e. self-blaming biases (right superior anterior temporal-subgenual connectivity), negative perceptual biases (amygdala) and dysfunction of task-independent subgenual networks. If these neural signatures were to relate to specific subtypes rather than independently predicting the same underlying general pathophysiology, then this would offer the intriguing possibility of stratification for neuromodulation and neurofeedback studies based on distinct neural circuits of interest, by either modulating self-blaming or emotional perception biases in patients non-responsive to standard treatments. The feasibility of such interventions has recently been confirmed, with reports of a training-induced reduction in self-blame-selective connectivity (Jaeckle et al. 2023) and an enhancement of amygdala responsiveness to positive autobiographical memories (Young et al. 2019).

However, it is important to first determine whether these neural signatures represent a trait-like feature of a fully remitting subtype of MDD, or whether it is also modulated by depressive state. For example, both self-blame-related and emotional perception-related changes have been identified in remitted MDD (Joormann and Gotlib 2007; Lythe et al. 2022; Ruhe et al. 2019). It is unclear whether these changes are more pronounced when people develop a recurrent episode or are merely due to underlying vulnerabilities which are not modulated by symptomatic state. This question is key to a deeper pathophysiological understanding of MDD in that little is known about how trait-related changes interact with precipitating biological and psychological trigger events to result in a depressive brain state, and how it affects subsequent response to treatment.

### Limitations

Due to our relatively modest sample size, we were unable to use cross-validated and data-driven machine learning algorithms, which may have improved the prediction model performance. Moreover, our sample consisted of chronic MDD patients, often with anxious distress and other co-morbidities. In addition, treatment was not standardised and, unlike previous studies in randomised controlled trials, did not allow us to distinguish spontaneous remission and placebo effects from treatment-related

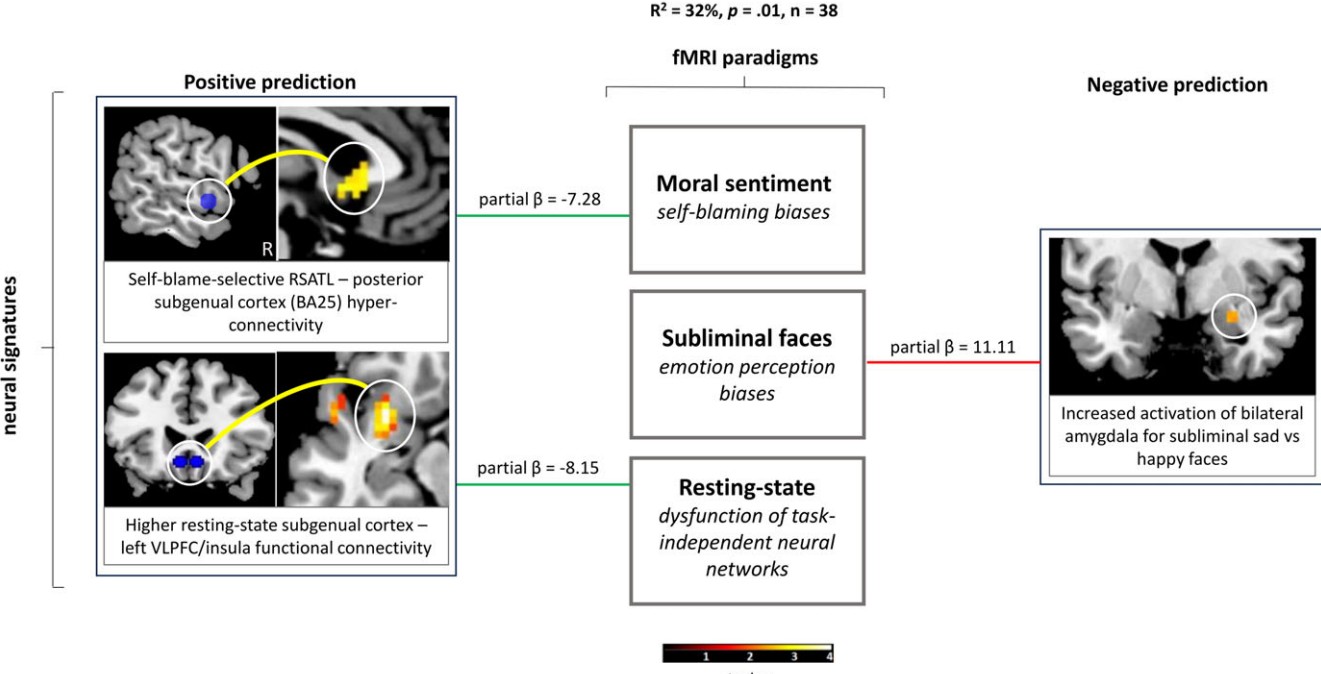

**Figure 1.** Neural signatures of emotional biases associated with clinical outcomes in difficult-to-treat MDD. Three neural signatures of emotional biases were associated with clinical outcomes in UK primary care. More specifically, it shows cropped sections of voxel-based analyses illustrating the respective pre-registered *a priori* regions-of-interest, i.e. self-blame-selective right superior anterior temporal lobe-posterior subgenual cortex (BA25) connectivity, resting-state functional connectivity between the subgenual cortex and ventrolateral prefrontal cortex/insula, and bilateral amygdala blood-oxygen-level-dependent activation in response to subliminal sad vs happy faces. These cropped sections are displayed using MRIcron at an uncorrected voxel-level threshold of *p*=.005, with no cluster-size threshold (the colour bar represents *t* values) and adapted from figures previously published (Fennema et al. 2023; Fennema, Barker, O'Daly, Duan, Carr, et al. 2024; Fennema, Barker, O'Daly, Duan, Godlewska, et al. 2024). A linear model using the pre-registered fMRI measures with baseline Maudsley Modified Patient Health Questionnaire (9 items) as a covariate explained 32% of the variance of QIDS-SR16 percentage change. The red and green lines display the partial effects of the fMRI measures on the variance of QIDS-SR16 percentage change after four months of standard primary care. MDD = major depressive disorder; BA = Brodmann Area; RSATL = right superior anterior temporal lobe; VLPFC = ventrolateral prefrontal cortex; BOLD = blood-oxygen level-dependent; QIDS-SR16 = Quick Inventory of Depressive Symptomatology, self-rated (16 items).

effects. Given the selection biases in randomised controlled trials, however, it was important to investigate a pragmatic sample as we have undertaken in this study.

Clinical utility is complicated by the heterogenous nature of MDD, resulting in patients with a wide variety of symptoms, disease severity and treatment history (Strawbridge et al. 2017), as well as patient response to treatment (Mayberg and Dunlop 2023). Further complementary predictive measures, such as novel cognitive markers (Lawrence et al. 2022), would be useful in addition to imaging markers to achieve clinically relevant levels of individual prediction of response to specific types of treatment.

Moreover, it is important to acknowledge that percentage-based reduction scores to define treatment response have been criticised, as it is biased towards more severe depressive symptoms at baseline (Rost et al. 2022). As a result, it is plausible for a responder to still experience clinically significant distress or impairment when starting with a baseline score in the severe range, while a non-responder may show a clinically significant improvement – which was also observed in the current study.

## Conclusions

Taken together, we reproduced clinically relevant neural signatures in an independent, pragmatic sample of difficult-to-treat MDD.

The findings confirm the pathophysiological relevance and potential of the proposed candidate neural signatures to make relevant contributions to the prospective prediction of clinical outcomes in more chronic, difficult-to-treat forms of MDD and call for stratified neurofeedback and neuromodulation interventions.

**Supplementary material.** The supplementary material for this article can be found at https://doi.org/10.1017/dep.2024.6.

**Data availability statement.** The data that support the findings of this study are available on request from the corresponding author, RZ. We will only be able to share fully anonymised, no pseudonymised data and requests will have to go through a King's College London repository.

**Acknowledgements.** We are grateful to Drs Mark Ashworth and Barbara Barrett who contributed to the trial study design, and to Drs Phillippa Harrison and Suqian Duan who collected trial data. We also thank the study participants for their support.

Part of the study has been published in a PhD thesis available on the King's College London institutional repository, Pure, see Fennema (2022): https://kclpure.kcl.ac.uk/portal/en/studentTheses/neural-signatures-of-emotional-biases-and-prognosisin-treatment.

**Author contribution. Diede Fennema:** Conceptualisation, Methodology, Formal analysis, Investigation, Data curation, Writing – original draft, Visualisation, Funding acquisition. **Gareth Barker:** Conceptualisation,

Methodology, Writing – review & editing, Supervision. **Owen O'Daly:** Conceptualisation, Methodology, Writing – review & editing. **Beata Godlewska:** Methodology, Resources, Writing – review & editing. **Ewan Carr:** Conceptualisation, Methodology, Writing – review & editing. **Kimberley Goldsmith:** Conceptualisation, Methodology, Writing – review & editing. **Allan Young:** Conceptualisation, Methodology, Writing – review & editing, Supervision, Project administration, Funding acquisition. **Jorge Moll:** Conceptualisation, Writing – review & editing. **Roland Zahn:** Conceptualisation, Methodology, Formal analysis, Writing – review & editing, Supervision, Project administration, Funding acquisition.

**Financial support.** This study represents independent research funded by the National Institute for Health and Care Research (NIHR) under its Research for Patient Benefit (RfPB) Programme (Grant Reference Number PB-PG-0416-20039) and independent research part funded by the National Institute for Health and Care (NIHR) Biomedical Research Centre at South London and Maudsley National Health Service (NHS) Foundation Trust and King's College London (Profs Zahn, Young, Goldsmith; Dr Carr). Prof Zahn was also partly funded by a Medical Research Council grant (ref. MR/T017538/1), while Prof Goldsmith was also supported by the National Institute for Health and Care Research (NIHR) Applied Research Collaboration South London (NIHR ARC South London) at King's College Hospital NHS Foundation Trust. This study was also supported by the Rosetrees Trust (M816) awarded to Prof Zahn. Dr Fennema was funded by a Medical Research Council Doctoral Training Partnership Studentship (ref. 2064430) and partly supported by a King's College London/D'Or Institute for Research and Education (KCL/IDOR) Pioneer Science Fellowship, funded by Scients Institute and the IDOR Pioneer Science Initiative. The views expressed are those of the authors and not necessarily those of the NHS, the NIHR or the Department of Health and Social Care. Additional support was provided to the study by the South London Clinical Research Network and sponsorship by Lambeth CCG.

**Competing interests.** Prof Zahn is a private psychiatrist service provider at The London Depression Institute and co-investigator on a Livanova-funded observational study of Vagus Nerve Stimulation (VNS) for Depression. Prof Zahn has received honoraria for talks at medical symposia sponsored by Lundbeck as well as Janssen. Prof Zahn has collaborated with EMOTRA, EMIS PLC and Depsee Ltd. Prof Zahn is affiliated with the D'Or Institute of Research and Education, Rio de Janeiro and advises the Scients Institute, USA. Prof Barker receives honoraria for teaching from GE Healthcare. Prof Young is employed by King's College London as an honorary consultant in the South London and Maudsley Trust (NHS UK) and is a consultant to Johnson & Johnson and Livanova. Prof Young has given paid lectures and sat on advisory open access boards for the following companies with drugs used in affective and related disorders: Astrazenaca, Eli Lilly, Lundbeck, Sunovion, Servier, Livanova, Janssen, Allegan, Bionomics, Sumitomo Dainippon Pharma. Prof Young has received honoraria for attending advisory boards and presenting talks at meetings organised by LivaNova. Prof Young is the Principal Investigator of the following studies: Restore-Life VNS registry study funded by LivaNova, ESKETINTRD3004: 'An Open-label, Long-term, Safety and Efficacy Study of Intranasal Esketamine in Treatment-resistant Depression', 'The Effects of Psilocybin on Cognitive Function in Healthy Participants' and 'The Safety and Efficacy of Psilocybin in Participants with Treatment-Resistant Depression (P-TRD)'. Prof Young has received grant funding (past and present) from the following: National Institute of Mental Health (USA); Canadian Institutes for Health Research (Canada); National Alliance for Research on Schizophrenia and Depression (USA); Stanley Medical Research Institute (USA); Medical Research Council (UK); Wellcome Trust (UK); Royal College of Physicians (Edin); British Medical Assocation (UK); University of British Columbia-Vancouver General Hospital Foundation (Canada); Wisconsin Economic Development Corporation (Canada); Coast Capital Savings Depression Research Fund (Canada); Michael Smith Foundation for Health Research (Canada); NIHR (UK); Janssen (UK). Prof Young has no share-holdings in pharmaceutical companies. Prof Goldsmith reports grants from NIHR, Stroke association, National Institutes of Health (US) and Juvenile Diabetes Research Foundation (US) during the conduct of the study. None of the other authors reports biomedical financial interests or potential conflicts of interest related to the subject of this paper.

**Ethical standards.** Ethical approval was obtained from the NHS Health Research Authority and National Research Ethics Service London – Camberwell St Giles Committee (REC reference: 17/LO/2074). Written informed consent was obtained from all participants.

## Connections references

Hickie I and Williams L (2023) Will new brain circuit focused methods (EEG, fMRI etc) lead to more personalised care options? *Research Directions: Depression* **1**(E12). https://doi.org/10.1017/dep.2023.22.

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
