## [Reviewer Report]

The manuscript by Fennema and colleagues investigated the utility of neural markers in predicting treatment outcomes for difficult-to-treat major depressive disorder (MDD). Specifically, they identified that in contrast to baseline symptom severity, which only explained 3% of outcome variance, three specific neural measures across three separate tasks explained 32%. These findings highlight the potential of these neural signatures as predictors of clinical outcomes in chronic, difficult-to-treat MDD. Despite the relatively small sample size and that individually these effects have been previously published, I do believe that there is some novelty in the approach taken in this study. Attached are my concerns in detail, I hope the authors find these comments helpful.

Introduction

The introduction is very short and does not adequately introduce the background for the topic, particularly for a journal which does not specialise in neuroimaging. While the study’s pre-registered regions of interest are a strength, engaging in wider and more recent literature concerning general prognosis or prediction of treatment outcome for SSRIs would be useful.

Line 52 “Despite these promising findings, their reproducibility has not been established and it is unclear whether these neural signatures generalise to pragmatic samples of patients encountered in clinical settings.” I don’t believe that this is the primary issue that this paper is set up to answer (given that apart from the amygdala the individual markers don’t replicate). It appears to be more tailored to examine whether together these parameters provide greater utility than was observed in the individual studies.

Methods

It wasn’t immediately clear from the manuscript that the descriptions of your tasks were located in the supplementary materials. A brief description of the tasks in the main manuscript and reference to the full description in the supplementary materials is warranted.

The pre-registration report mentions a number of additional regions of interest including functional connectivity between SCC - VMPFC and SCC - dorsal midbrain, and pregenual ACC activity for the implicit face processing task as a region of interest which do not appear in this study. Why were these not included in the analysis model? Also, it highlights that a logistic regression will be used to predicted binarised response/non-response. While I think that symptom change is a more useful measure, I believe that examining the binarised outcome as well would also be informative to readers (and consistent with your power analysis).

“baseline Maudsley Modified Patient Health Questionnaire, 9 items 101 (MM-PHQ-9; measure of severity of depressive symptoms) (Harrison et al. 2021) as a covariate”. Why wasn’t baseline QIDS used as the covariate here? Surely this would do a better job of capturing the same depressive symptoms at baseline.

Results

An additional table examining any baseline differences between responders/non-responders at baseline would be useful in identifying whether any factors were confounding the prediction results.

Discussion

The discussion does not adequately articulate this study’s addition to the literature and the author’s interpretation of the findings. Given that they have used measures of activity and connectivity which have been previously published, it is important to emphasise the novelty of this paper in combining these measures. It is also important, given the non-specific nature of the treatment being applied and the general prognostic effects identified, that the hypothesised clinical utility of such findings are explicitly stated and interpreted (e.g. identifying a prognostic marker no matter how good does not eliminate the trail-and-error nature of prescribing antidepressants).

Furthermore, it is important for the authors to highlight how they suggest to improve the explained variance in future studies. While 32% is much better than the amount given by the clinical variables alone, it is unlikely sufficient for translation given the costs associated with running three different scans.

Line 174 “This offers the intriguing possibility of stratification for neuromodulation and neurofeedback studies based on distinct neural circuits of interest, by either modulating self-blaming or emotional perception biases in patients non-responsive to standard treatments”. While I agree with this statement generally, the fact that the measures were uncorrelated isn’t evidence for this point. It is possible that all these features independently predict the same underlying general pathophysiology. Without testing whether these features relate to specific symptoms this point is therefore difficult to disentangle.

Minor notes

Line 36 “to respond more strongly” please be specific to what you mean by strongly here.

Your reference manager appears to have formatted the in-text citations strangely.

## Score Card

### Presentation

3.0/5

Is the article written in clear and proper English?
30%
4/5

Is the data presented in the most useful manner?
40%
3/5

Does the paper cite relevant and related articles appropriately?
30%
3/5

### Context

3.0/5

Does the title suitably represent the article?
25%
4/5

Does the abstract correctly embody the content of the article?
25%
4/5

Does the introduction give appropriate context and indicate the relevance of the results to the question or hypothesis under consideration?
25%
2/5

Is the objective of the experiment clearly defined?
25%
2/5

### Results

3.0/5

Is sufficient detail provided to allow replication of the study?
50%
3/5

Are the limitations of the experiment as well as the contributions of the results clearly outlined?
50%
3/5

---

## [Reviewer Report]

1. Justify the use of %change in self-reported QIDS as main clinical outcome - %change has known limitations in outcomes research compared with other absolute change measures, and thresholds for absolute improvement; As part of this, additional reporting of the distribution of absolute changes in QIDS scores would be helpful in the actual text and not just supplementary tables.

2. Some further elaboration of the proposed (structural or functional) circuitry delineated here – not just in terms of the propsed ‘cogntive aspects (‘self-blame’, ‘negative perception’) but their actual anatomical or physiological characteristics and the extent to which they related to circuits proposed by others (notably Williams et al) for predicting depression outcomes on medication.

## Score Card

### Presentation

3.0/5

Is the article written in clear and proper English?
30%
4/5

Is the data presented in the most useful manner?
40%
3/5

Does the paper cite relevant and related articles appropriately?
30%
3/5

### Context

3.0/5

Does the title suitably represent the article?
25%
4/5

Does the abstract correctly embody the content of the article?
25%
4/5

Does the introduction give appropriate context and indicate the relevance of the results to the question or hypothesis under consideration?
25%
2/5

Is the objective of the experiment clearly defined?
25%
2/5

### Results

4.0/5

Is sufficient detail provided to allow replication of the study?
50%
4/5

Are the limitations of the experiment as well as the contributions of the results clearly outlined?
50%
4/5

---

## [Editor Report]

The authors have provided a thoughtful revision of the manuscript and have appropriately addressed the reviewers’s comments. I am happy to recommend that this manuscript be Accepted.